# Environmental Pollution Effect Analysis of Lead Compounds in China Based on Life Cycle

**DOI:** 10.3390/ijerph17072184

**Published:** 2020-03-25

**Authors:** Jianbo Yang, Xin Li, Zehui Xiong, Minxi Wang, Qunyi Liu

**Affiliations:** 1China Enfei Engineering Corporation, Beijing 100011, China; yangjianbo@cags.ac.cn; 2Chengdu University of Technology, Chengdu 610059, China; xiongzehui2019@sina.com; 3Chinese Academy Geological Sciences, Institute Mineral Resources, Beijing 100037, China; liuqunyi@cags.ac.cn

**Keywords:** lead pollution, material flow analysis, emission inventory accounting, contamination control

## Abstract

Environmental pollution caused by lead toxicity causes harm to human health. Lead pollution in the environment mainly comes from the processes of mining, processing, production, use, and recovery of lead. China is the world’s largest producer and consumer of refined lead. In this paper, the material flow analysis method is used to analyze the flow and direction of lead loss in four stages of lead production, manufacturing, use, and waste management in China from 1949 to 2017. The proportion coefficient of lead compounds in each stage of lead loss was determined. The categories and quantities of lead compounds discharged in each stage were calculated. The results show that in 2017, China emitted 2.1519 million tons of lead compounds. In the four stages of production, manufacturing, use, and waste management, 137.9 kilo tons, 209 kilo tons, 275 kilo tons, and 1.53 million tons were respectively discharged. The emissions in the production stage are PbS, PbO, PbSO_4_, PbO_2_, Pb_2_O_3_, and more. The emissions during the manufacturing phase are Pb, PbO, PbSO_4_, Pb_2_O_3_, Pb_3_O_4_, and more. The main emissions are Pb, PbO, Pb_2_O_3_, Pb_3_O_4_, and more. The main emissions in the waste management stage are PbS, Pb, PbO, PbSO_4_, PbO_2_, PbCO_3_, Pb_2_O_3_, Pb_3_O_4_, and more. Among them, the emissions of PbSO_4_, PbO, Pb, and PbO_2_ account for about 90%, which are the main environmental pollution emissions. The waste management stage is an important control source of lead compound emission and pollution. In view of these characteristics of the environmental pollution risk of lead compounds in China, the government should issue more targeted policies to control lead pollution.

## 1. Introduction

Lead (Pb) is a naturally occurring element found in the earth’s crust with various uses, largely owing to its malleability and corrosion resistance. Multiple uses of Pb during the 20th century, including leaded petrol, lead-based paint, and solder in water pipes, have resulted in elevated Pb exposure risk to humans. Young children, in particular, are more vulnerable to Pb exposure due to their frequent hand-to-mouth activity, higher absorption rates, and their developing central nervous system [1]. As a newly industrialized country, China’s demand for mineral resources and per capita consumption have increased year by year [2]. In 2017, China’s refined lead consumption reached 4.723 million tons, accounting for 42% of global consumption [3]. While satisfying the consumption of mineral resources, China has also produced a large amount of waste. This waste increases by 10% every year [4]. The waste contains rich mineral resources, and the recycling of resources slows down the rate of natural mineral depletion [5]. Therefore, the analysis of metal resource material circulation or economic metabolism has gradually become a research hotspot. Luca et al. (2011) clearly depicts the whole process of production, consumption, social accumulation, and recycling of China’s aluminum metal through material flow analysis, providing support for the Chinese government to formulate resource policies [6]. It can also provide quantitative scientific methods for promoting cleaner production and a circular economy [7]. In addition, material flow analysis is also widely used in the analysis of the recycling efficiency of lead and hazardous material flow networks [8]. This can help humans to identify and predict the flow and direction of specific substances, and scientifically analyze the efficiency of human resources utilization. The higher the efficiency, the lesser the impact on the environment, and the better the human control ability [9].

Lead is a type of corrosion-resistant heavy nonferrous metal material. It is widely used in the chemical industry, and for cable, battery, and radiation protection. Because lead is a widespread urban pollutant, it has been strictly controlled around the world [10]. According to the World Health Organization, lead exposure contributes to about 600,000 new cases of children with intellectual disabilities every year. Aside from the poisoned futures these children suffer, the economic losses are huge: by lowering the IQ of children, lead exposure costs low- and middle-income countries $977 billion per year [11]. China is the world’s largest producer and consumer of refined lead [12]. The effective control of lead production, consumption, use, and recycling is an important prerequisite to reduce lead exposure. The present study found a geometric mean value blood lead level (BLL) of 26.7μg/L in Chinese children, with 8.6% exceeding 50μg/L. However, compared to countries with a very high Human Development Index (HDI ≥ 0.9) such as Japan, Australia, France, Canada, and the US, Chinese children have much higher BLLs [13]. Lead poisoning has occurred in many regions of China, such as Anhui Province, Jiangsu Province, Fujian Province, Sichuan Province, Yunnan Province, and Guangdong Province [14]. Some studies have shown that the content of lead in the main heavy metals of China’s municipal solid waste is relatively high, resulting in a relatively high content of lead in the soil near the municipal solid waste incinerator and in the residence [15]. Some studies also show that the lead content in industrial wastewater and urban wastewater exceeds the standard [16]. On July 23, 2019, lead was listed in the list of toxic and harmful water pollutants by China’s Environmental Ecology Department [17]. At the same time, lead is contained in house decoration materials, children’s toys, stationery, cosmetics, batteries, automobile exhaust, and factory dust to varying degrees. Lead batteries are the main consumption field of lead in China, and their consumption accounts for 80% of the total consumption [18]. Lead batteries are the main control type of China’s renewable lead industry. In April 2019, the Chinese government promulgated the technical specification for the recovery of waste lead–acid batteries, which specifies the main responsibility and pollution prevention measures of lead–acid batteries in the process of "production recovery disposal". In 2014, “The Bulletin of China’s Soil Pollution Investigation” also showed that the value of lead in cultivated land over the national standard rate was 1.5%, including lead in the main pollutants in the metal smelting industrial park and its surrounding soil [19].

The loss of lead in the whole life cycle of production, consumption, use, and recycling is mainly in the form of tailings, solid waste, dross, ash, fly ash, sludge, erosion, and other forms. The main substance forms are Pb, PbO, Pb_3_O_4_, PbCO_3_, PbCl_2_, PbO_2_, Pb_2_O_3_, PbSO_4_, and others [20]. These lead compounds are lost to soil, water, and air and cause lead pollution. Exposure to the environment that humans can touch can also lead to lead poisoning. Since the 21st session of the United Nations Environment Program in 2001, a series of resolutions on lead control have been formed to promote global action to control the environmental exposure and health risk of lead. China’s continuous measures are effective, for example, in the areas of mining, metal smelting, and production, lead emissions and environmental impact have declined significantly. The continuous decline of the blood lead index in children is important proof of this. The focus of lead pollution has gradually shifted to the use and scrap management of urban lead products. The existing research results show that the distribution of the blood lead index is unbalanced in all provinces of China [13]. This requires a new perspective to assess the environmental impact of lead metabolism. Material flow analysis is a systematic evaluation tool which can characterize the flow and flow dynamic changes of lead and its compounds in life cycle. This allows us to grasp the impact of lead on the environment in the whole life cycle and assess the health risk, and helps us to identify the direction of environmental management of urban lead and other hazardous products. It also provides us with the means to decide on support for the sustainable development of urban health. Therefore, this paper takes lead as the object, and calculates the lead logistics volume, flow direction, and loss in each stage (production, consumption, use, and recovery) of lead in China from 1949 to 2017 through the material flow analysis, and calculates the list of environmental emissions in 2017, which provides corresponding support for the objective analysis of lead environmental pollution risk and the government’s lead pollution control policy.

## 2. System Boundary and Methods

### 2.1. MFA and System Boundary

Material flow analysis (MFA) refers to the systematic analysis or evaluation of a specific material flow and storage in a certain space–time range, which is one of the methods for industrial metabolism research. Material flow analysis links resources, path, intermediate process, and final destination together, and follows the law of conservation of mass. It evaluates the relationship between economic growth and material flow through quantitative analysis of material input and output in social and economic activities and establishes quantitative relationships between material input and output [21].

Based on the material flow method, the lead life cycle is divided into production, manufacturing, use, waste management, and other stages. The loss of lead according to the flow and direction of substances in each stage is calculated. Then, according to the category proportion of lead compounds lost in each stage, the emissions of Pb, PbO, PbO_2_, PbS, PbSO_4_, Pb_2_O_3_, Pb_3_O_4_ and other major lead emissions are estimated to form a material flow diagram of the main emissions of lead in the life cycle. This can provide data and method support for lead control in all stages. According to the above research process and content, the system boundary of this paper is formed, as shown in Figure 1.

### 2.2. Material Flow Calculation

Material flow analysis (MFA) is based on the conservation of material. The sum of the material output in each stage should be equal to the sum of input, as shown in Equation (1):(1)$$F^{\mathrm{input}}+F^{\mathrm{import}}=F^{\mathrm{output}}+F^{\mathrm{export}}+F^{\mathrm{loss}}$$
where, *F*^input^; *F*^import^; *F*^output^; *F*^export^; *F*^loss^ respectively refer to the material input, import, output, export, and loss of lead material flow at a certain stage or the whole national interface. In this paper, the lead material flow rate is lead metal content.

### 2.3. Trade Flow Calculation

The lead material flow in the process of import and export trade is calculated according to the physical quantity of lead containing commodities and its lead coefficient. For the calculation formula of lead content in import and export commodities, see Equations (2) and (3).
(2)$$M_{p}=\overset{n}{\underset{p=1}{\sum }}C_{p}\times R_{p}$$
(3)$$E_{p}=\overset{n}{\underset{p=1}{\sum }}X_{p}\times R_{p}$$
where *p* represents the different types of lead-containing commodities in the import and export trade; *Cp* and *Xp* represent the physical quantity of lead imports and exports of *p*-type lead; *Mp* and *Ep* represent the lead content of imported and exported lead-containing commodities in category *p*, respectively.

### 2.4. Lead Content Coefficient

This article determines its lead content based on its material composition. There are many types of lead-containing product data, and this study uses stratified sampling statistical methods to process these data. In this study, the relevant standards, material composition, and other parameters were consulted. In combination with the product parameters of various related industries, different types of lead-containing coefficients were set for different lead-containing end products. In this paper, the tons are metric tons (1000 kg), as shown in Table 1.

On the basis of determining the lead content of each small category of leaded end products, a weighted average calculation is performed to obtain the lead content of each major category of leaded end products, as shown in Table 2.

### 2.5. Calculation of Lead Loss

According to the main economic and technical indicators in the "Compilation of China Nonferrous Industry Association", the material loss of lead is calculated. This study includes the calculation of lead material loss in the smelting, processing and manufacturing, use, and waste recycling stages.

In the production stage, the lead loss includes three aspects: the tailings loss of lead ore in the process of beneficiation; the lead loss of primary lead in the process of blast furnace sintering and electrolytic refining; the lead loss of regenerated lead in the smelting process of the reverberatory furnace. Therefore, the loss of ore is calculated in Equation (4).
(4)$$L=L^{\prime }+L^{\prime \prime }+L^{\prime \prime \prime }$$
where *L*′ is the lead material loss during the beneficiation process, *L*″ is the lead material loss during the smelting process of the lead concentrate, and *L*‴ is the lead material loss during the electrolytic process of the crude lead. See Equations (5-7) for calculation.
(5)$$L^{\prime }=P\times (1-\gamma_{1})$$
(6)$$L^{\prime \prime }=\left(P\times \gamma_{1}+I\right)\times \left(1-\gamma_{2}\right)$$
(7)$$L^{\prime \prime \prime }=\left(P\times \gamma_{1}+I\right)\times \gamma_{2}\times \left(1-\gamma_{3}\right)$$
where $\gamma_{1}$ is the recovery rate of lead ore in beneficiation; $\gamma_{2}$ is the recovery rate of primary lead in smelting process; $\gamma_{3}$ is the recovery rate of recycled lead smelting process; *P* is the lead content of domestic lead ore production; *I* is the lead content of imported lead concentrate.

The lead losses in the above research links include the direct environmental emissions and the waste containing lead to be treated. Considering that the direct environmental emissions in actual production are very few and the utilization rate of the waste containing lead is very low, the direct environmental emissions and the waste containing lead to be treated are counted as the lead material losses in this study, and are not divided into the dispersed losses and inventory losses.

### 2.6. Calculation of Lead Compounds Emission Ratio

The chemical forms of lead include two aspects: species and quantity distribution. Among them, the speciation includes the occurrence state and chemical structure of elements [22]. In this study, the chemical structure of environmental release of lead elements is mainly discussed, and the complex combination of lead and other substances is not considered. For example, PbO is not only contained in smelting lead slag, but also combined with other substances as the composition of lead paste in lead battery. No matter what the combination status, the lead contained in lead slag and lead paste is the same chemical structure. In terms of quantity distribution, the quantity involved in this paper refers to the lead content in the lead release. At some stage of the life cycle, the amount of lead released by form *Y* is:(8)$$Y=\overset{n}{\underset{I=1}{\sum }}R_{i}X_{i}$$
where *i* represents a process in the lead metal production stage (including mining and beneficiation, primary lead smelting, and recycled lead smelting) or other special lead products (such as lead–acid batteries, cable sheathes, etc.) in three stages of the life cycle. *n* is the total number of production processes in the production stage, or the total number of lead product categories in other stages. *X_i_* is the lead discharge corresponding to the *i* th process in this stage, *R_i_*
*i*s the emission structure coefficient.

## 3. Results

### 3.1. Production Stage

#### 3.1.1. Production of Primary and Secondary Lead

The lead used by human beings originally came from the lead ore mined in mines. Lead ore concentrate with a lead content of about 60% is formed after beneficiation. Lead concentrate is smelted to produce lead metal with a purity greater than 99.97%, which is called refined lead. Among them, the refined lead produced from lead concentrate is called primary lead, and the refined lead produced from lead containing waste is called recycled lead. Figure 2 shows the production changes of primary and secondary lead in China from 1949 to 2017. In 2017, China’s production of primary lead and recycled lead was 2.677 million tons and 2.046 million tons, respectively.

#### 3.1.2. Calculation of Lead Loss

The grade of lead concentrate is about 60%, and the lead concentrate is sent to the smelting furnace to produce the crude lead containing about 95% lead. The recovery rate of China’s crude lead smelting increased from 88.2% in 1949 to 97.8% in 2017. The crude lead is further refined into high-purity electrolytic refined lead with more than 99.99% lead. The recovery rate of lead electrolysis also increased from 90.7% in 1949 to 97.2% in 2017. 

According to the domestic output, import volume, and recovery rate of lead smelting of China’s lead concentrate, it is calculated that the loss of lead in the primary lead smelting stage from 1949 to 2017 was 2.95 million tons. Among them, the loss in 2017 was 77 kilo tons. The loss of lead in the smelting stage of regenerated lead is 0.850 million tons. Among them, the loss in 2017 was 60 kilo tons, as shown in Figure 3.

#### 3.1.3. Inventory Accounting of Lead Loss

China’s primary lead smelting mainly adopts the sintering blast furnace process [24], and then uses the Betts electric process to refine more than 99% of the refined lead. The slag contains lead oxide (PbO, Pb_2_O_3_, Pb_3_O_4_), sulfide (PbS), and a small amount of PbSO_4_. During electrolysis, anode slime containing PbO and PbFeCl, dross containing PbS, and oxide slag containing PbO are formed. The proportion of PbO, PbS, PbSO_4_, Pb_2_O_3_, Pb_3_O_4_, and others are 40%, 30%, 12%, 10%, 6%, and 2%, respectively.

The reverberatory furnace [25] is used for the smelting of recycled lead in China, and the waste lead battery accounts for 85% of the raw materials [26]. PbSO_4_, PbO_2_, Pb, PbO, and other lead-containing substances are adhered on the waste battery shell and partition board containing lead [27]. In the smelting process, lead filings, soda, and crushed coke are added for reduction, sulfur fixation, and slagging. The smelting lead slag contains PbS, PbO, PbSO_4_, and others, and the lead dust contains PbS, PbO, and others. According to the lead balance table [28], we can observe that the proportion of PbS, PbO, PbSO_4_, PbO_2_, and others is 34.6%, 22.0%, 18.4%, 23.3%, and 1.7%, respectively.

The loss of various lead compounds during the lead smelting stage in China in 2017 is shown in Table 3.

### 3.2. Manufacture Stage

#### 3.2.1. Consumption Structure of Refined Lead

Lead products include lead–acid batteries, lead compounds, coatings and additives, solders, cable sheaths, lead bullets, and others. Therefore, Figure 4 describes the consumption structure change of refined lead in China from 1949 to 2017. Although lead has many uses, it is mainly concentrated in lead–acid batteries. In 2017, the consumption of lead–acid batteries accounted for 73.5% of lead products.

#### 3.2.2. Calculation of Lead Loss

In this paper, lead loss is estimated by referring to the national standard of the China Emission Standard of Pollutants for Battery Industry (GB 30484-2013). The lead emission limit of water pollution unit production in the process of plate manufacturing and assembly of existing lead–acid battery enterprises is 0.2 kg/kVAh. Compared with the lead content of 20 kg/kVAh, the loss rate is only 1%. The production process of lead–acid batteries is divided into lead powder and grid manufacturing, paste mixing, curing assembly, and other processes. The waste containing lead in the processing and manufacturing is mainly lead alloy and lead paste. The comprehensive loss rate of PbSb alloy (95% Pb) and PbAl alloy (96% Pb) is 2%. Lead Oxide and lead salt are mainly used in stabilizers, lead paste of lead–acid batteries, lead glass, and others. The lead loss rate in the process is 1%. According to the calculation, the cumulative loss of lead in China’s processing and manufacturing stage from 1949 to 2017 was 3.69 million tons from leadؘ–acid batteries, 210 kilo tons from 1lead alloy, and 120 kilo tons from lead oxide, respectively. 

As shown in Figure 5, in 2017, the lead losses in China’s processing and manufacturing stage were 201 kilo tons from lead–acid batteries, 6.7 kilo tons from lead alloys, and 14 kilo tons from lead oxides, respectively.

#### 3.2.3. Inventory Accounting of Lead Loss

The main lead alloy lost in the processing of lead–acid battery modules is lead antimony alloy (Pb_x_Sb_y_) or lead aluminum alloy (Pb_x_Al_y_). The composition of lead paste is PbO, 3PbO · PbSO_4_ · H_2_O and PbO · PbSO_4_ [29]. The proportion of lead alloy and paste waste produced in manufacturing is about 1/2 [30]. As a result, the proportion of Pb, PbO, and PbSO_4_ lost in this stage is about 33.4%, 53.3%, and 13.3%, respectively.

The surface of molten lead is covered with oxide dross and its composition is PbO. At the same time, a small amount of lead vapor and lead dust is produced. The lead slag contains CaPb_3_ and a small amount of lead metal. The loss ratio of PbO, Pb, and others in lead alloy smelting is 85%, 10%, and 5%, respectively. The loss list of lead compounds in China’s lead manufacturing stage in 2017 is shown in Table 4.

### 3.3. Use Stage

#### 3.3.1. Main Categories of Lead Products

Lead products mainly include lead–acid batteries, lead bullets, cable sheaths, solders, lead chemicals, and other products. The loss in the use stage mainly includes three aspects. The first is the loss of products, such as paint, ammunition, plastic additives, solder, and so forth. The lead in these products is released into the environment during their use. The second is the cable sheath, which is difficult to be recovered in situ or the cost is too high, and often remains underground or in the seabed. This lead is also known as “hibernating lead” [31]. The third is the environmental diffusion and corrosion caused by the weathering and wear of lead products [32].

#### 3.3.2. Lead Loss in Use Stage

For non-dispersive lead containing products, there is no lead loss and emission in the use process. It is difficult to recycle the lead of the lost lead products (such as lead chemicals, lead solder, cable sheathes, and lead bullets). This paper assumes that it is completely dissipated into the environment. From 1949 to 2017, the lead losses of lead chemicals, lead solder, cable sheaths, and lead bullets were 5.24 million tons, 1.81 million tons, 1.35 million tons, and 1.09 million tons, respectively, as shown in Figure 6. Among them, the losses were 237 kilo tons, 19 kilo tons, 14 kilo tons, and 5 kilo tons respectively in 2017.

#### 3.3.3. Inventory Accounting of Lead Loss

Lead products in the use stage will produce different forms of environmental emissions. Lead chemicals, lead oxides, cable sheaths, lead bullets, and other lead substances released to the environment in the process of use are often different. The average environmental release of PbO, Pb_3_O_4_, and Pb_2_O_3_ in lead chemicals is 65%, 25%, and 10%, respectively. Lead containing solder and additives are mostly lead salts, such as 2PbCO_3_ · Pb (OH)_2_, PbCl_2_, PbCrO_4_, etc. The loss ratio of Pb and PbCl_2_ is 25% and 75%, respectively. The loss ratio of Pb and PbO in cable sheathes is 95% and 5%, respectively. The main form of lead released into the environment is 100% Pb. Table 5 shows the list of lead compound emissions during the lead use phase in China in 2017.

### 3.4. Waste Management Stage

After the lead products reach the life cycle, they are gradually withdrawn from use and become waste. The management of leaded waste includes the recovery and subsequent treatment of waste lead acid batteries and other leaded waste. Waste treatment includes the collection and separation of waste products. Part of the lead contained in the waste can be recycled by separation and recovery. Other waste parts are sent to incinerators or landfills. Lead in the incineration process is lost or recovered along with the ash and fly ash produced by incineration. The higher the recovery efficiency in the lead waste management stage, the lower the lead loss and the lower the impact on the environment.

#### 3.4.1. Estimation of Scrap

In this study, the life cycle method was used to estimate the amount of lead waste in China from 2000 to 2017. Based on the data of the China renewable resources industry development report, this paper makes a comparative analysis. Then, the type and quantity of lead products in use inventory is determined. The estimated results are shown in Figure 7. From 2000 to 2017, 29.27 million tons of lead products were discarded in China. In 2017, China recycled 3.24 million tons of lead–acid batteries, 11.65 kilo tons of lead chemicals, and 18.1 kilo tons of solder and additives.

From 2000 to 2017, China has recovered 17.13 million tons of lead–acid batteries, 540 kilo tons of lead chemicals, and 1.04 million tons of solder and additives. As shown in Figure 8. In 2017, China recycled 1.866 million tons of lead–acid batteries, 70 kilo tons of lead chemicals, and 110 kilo tons of solder and additives.

#### 3.4.2. Inventory Accounting of Lead Loss

More than 80% of the lead-containing waste recovered in China is landfilled and the rest is incinerated. The types of leaded waste include municipal solid waste, construction waste, hazardous waste, and sewage sludge. The composition of waste lead–acid battery is different from that of new batteries. Among them, the proportion of PbSO_4_, PbO_2_, Pb, and PbO is 50%, 20%, 17%, and 13%, respectively [33]. The main components of the coating are Pb_3_O_4_ (Red lead) accounting for 25% and PbCO_3_ (White lead) accounting for 20%. The proportion of Pb_2_P_3_, PbO, PbSO_4_, and PbCl_2_ produced by incineration of the coating is 5%, 31.3%, 3.4%, and 15.3%, respectively. Lead chemical products in the waste disposal and recovery stage released are more complex. The proportion of PbO, Pb_2_O_3_, PbSO_4_, Pb_3_O_4_, and PbCl_2_ produced by incineration is 63.3%, 10%, 3.4%, 10%, and 13.3%, respectively. Table 6 shows the list of emissions of lead compounds in China’s lead waste management stage in 2017.

### 3.5. Results of Lead Cycle and Environmental Impact in China in 2017

Based on the above comprehensive quantitative analysis, the material flow diagram of four stages of lead surface cycle in China in 2017 was formed, as shown in Figure 9. In 2017, China’s refined lead production was 4.723 million tons. China imports pure lead, lead materials, waste lead, and other processing materials. In 2017, China consumed 3.78 million tons of lead products. The accumulated social stock of lead products in use was 22.95 million tons. In that year, 3.54 million tons of waste lead products were discarded, of which 2.046 million tons were recycled, with a recovery rate of 57.8%. In the production, consumption, use, and waste management stages, respectively, 137.9 kilo tons, 209 kilo tons, 275 kilo tons, and 1.53 million tons of lead compounds (all lead content) were discharged. In total, 2.519 million tons of lead compounds were emitted to the environment, a 60% reduction compared with 5.478 million tons in 2010 [34]. In 2017, the amount of environmental emissions per 1kg of refined lead consumed in China was 0.142kg. This result is far lower than the average level of 0.5kg lead compounds entering the environment for every 1kg of lead consumed in the world [35,36]. It can be seen that China is doing better and better in controlling the environmental impact of lead.

## 4. Discussion

The environmental emission of lead compounds is affected by metallurgy, production level, lead recycling efficiency, and consumption structure of lead products. In 2017, the proportion of recycled lead in China reached 42%, although it is still lower than the average recovery rate of 65% in developed countries. However, with the strengthening of China’s green development policy, the improvement of lead recovery technology, and the increase of economic feasibility, the efficiency of China’s lead recycling will continue to grow steadily. This will greatly reduce the amount of environmental releases of PbCO_3_, PbSO_4_, and PbS caused by primary lead production.

From the situation in 2017, the recovery rate of lead waste management in China is close to 60%, and the recovery efficiency still has room to improve. The emissions of lead compounds (such as PbSO_4_, PbO, Pb, PbO_2_) will decrease with the increase of recycling efficiency. Waste lead products (or municipal waste) will bring toxic lead pollution to the environment, whether burned or buried. Since 2013, China’s refined lead consumption has continued to decline. The substitution of lithium batteries for lead batteries has oversupplied China’s lead market. On the surface, the decrease of consumption will reduce the environmental impact of lead consumption. However, the continuous decline of lead’s price will lead to the lack of lead recovery power and the reduction of recovery efficiency. Environmental emissions of lead compounds may rebound. This needs to be considered by the government to make corresponding industrial support policies [34].

At present, from the perspective of reducing environmental impact, lead-freedom has become a trend. In the future, the application of dissipative lead products, such as cathode ray tubes, lead bombs, plastic stabilizers, and lead solder will be further reduced. At the same time, the application demand of lead in coatings, automotive oil, ordinary batteries, and daily necessities is still growing. If there is no green technological innovation, the release of PbO, Pb, PbSO_4_, and PbO_2_ in the production, use, and waste treatment stage of these lead products will remain at a static level. The government and industry should pay attention and enact controls appropriately.

From the four stages of the lead life cycle, the environmental impact of lead compounds under the constraints of a series of industrial control policies in China is moving towards the back end of the industrial chain. A large number of emissions are mainly concentrated in the use and waste management stage. The lead compounds in the production and manufacturing stage are mainly solid wastes, and waste residues in the processing process, which can continue to be improved through cleaner production. The lead compounds in the use stage are mainly caused by corrosion and wear. The environmental exposure stability of lead products should be monitored and supervised. The use of lead products that are in close contact with people should be reduced. In the lead waste management stage, the pollutant form of lead compounds can be observed as solid waste, fly ash, ash, and others. The standardized recovery and treatment of discarded lead products can reduce human exposure and lead exposure. The improvement of incinerator technology can improve the deposit recovery of incineration slag and fly ash, and reduce environmental pollution. It is very difficult to deal with lead pollution in sewage treatment, and there is also a risk of pollution in the composting of leaded waste. More attention and management should be applied to these observations.

## 5. Conclusions

(1) In 2017, the environmental emissions of lead compounds in four stages of the lead life cycle in China were 137.9 kilo tons, 209 kilo tons, 275 kilo tons, and 1.53 million tons, respectively. The majority of this was from the product use stage and waste disposal and recovery stage (84%).

(2) In 2017, the main form of environmental lead compounds produced in the lead life cycle was PbSO_4_, accounting for 23.4% of the total, followed by PbO, Pb and PbO_2_. These four kinds of lead compounds account for about 90% of the total emissions.

(3) PbO is mainly distributed in the production stage and waste management stage. The amount of PbSO_4_, Pb, and PbO_2_ distributed from the waste management stage is also the highest. The lead release of Pb_2_O_3_ and Pb3O4 mainly comes from the product use stage.

(4) PbO, Pb_2_O_3_, Pb_3_O_4_, PbO_2_, PbS, and PbSO_4_ are all lead compounds. They can enter the human body through the respiratory tract and the digestive tract, where there are human bones and the brain, liver, kidney, and other organs. It can cause serious diseases in the nervous system, digestive system, and hematopoietic system. In 2017, China’s total emissions of these major lead compounds exceeded two million tons. According to the data of the discharge and flow direction of lead compounds in the four stages of the study, the government should issue more targeted policies to control lead pollution.

(5) The research on the environmental impact of lead and other toxic metals should be changed from the national level to the regional level, especially the urban level. The results of this study show that the environmental impact of the lead treatment effect is significant. The focus of lead pollution control has changed from “flow” management of industrial production to “stock” of social consumption. The city is the accumulation center of social consumer goods. The use and discarding management of toxic metal products has become the main objective of urban environmental improvement and health management. Quantitative analysis of the geographical distribution of toxic metal products, as well as waste recycling and treatment of pollution, is of great significance to provide an improvement and a governance space for the urban environment and health management.

(6) The research methods and ideas of this paper can also be extended to the quantitative analysis of the environmental impact of heavy metals such as cadmium, mercury, copper, nickel, zinc, and manganese. Through collaborative and coupling research with urban water resources, the impact of heavy metal products on water, atmosphere, and soil in the process of using dissipation and waste incineration and landfill was analyzed. The analysis promotes strict “footprint” management of heavy metal products in cities, which has a very good supporting role for the comprehensive management and sustainable development of the urban environment.

## Figures and Tables

**Figure 1 ijerph-17-02184-f001:**
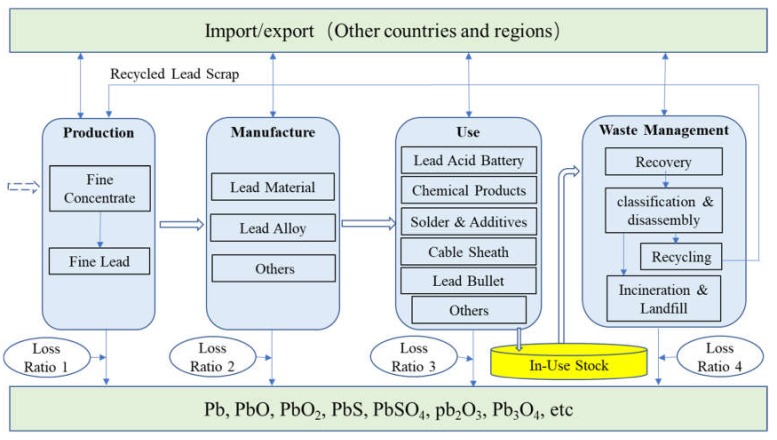
Material flow framework and system boundary of lead.

**Figure 2 ijerph-17-02184-f002:**
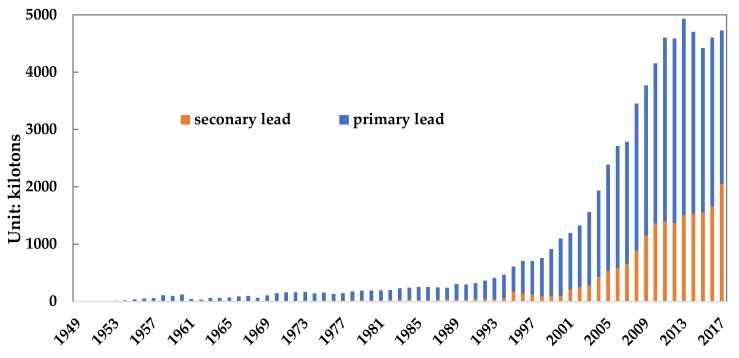
Changes in lead concentrate production from 1949 to 2017. (Data source: [22,23]).

**Figure 3 ijerph-17-02184-f003:**
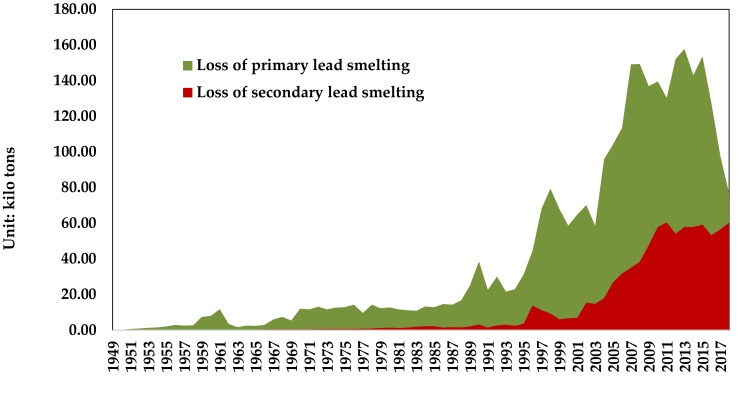
Lead material loss in lead smelting stage in China from 1949 to 2017. (Data source: [3,23]).

**Figure 4 ijerph-17-02184-f004:**
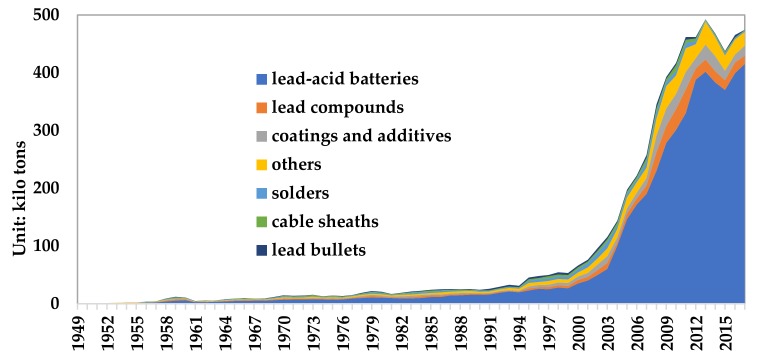
Consumption structure of refined lead from 1949 to 2017. (Data source: China Nonferrous Industry Association; Beijing Antaike Information Co., Ltd.).

**Figure 5 ijerph-17-02184-f005:**
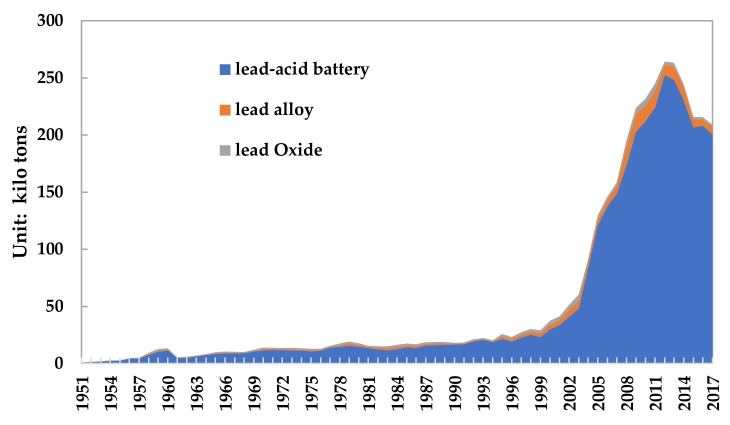
Lead material loss in the manufacturing stage of lead products from 1949 to 2017. (Data source: China Nonferrous Industry Association).

**Figure 6 ijerph-17-02184-f006:**
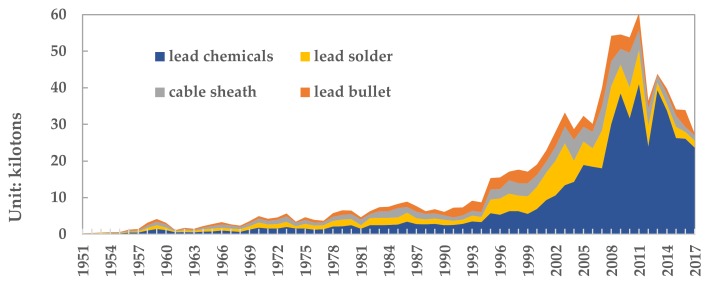
The loss of lead in the use stage of lead products from 1949 to 2017. (Data source: China Non-ferrous Industry Association).

**Figure 7 ijerph-17-02184-f007:**
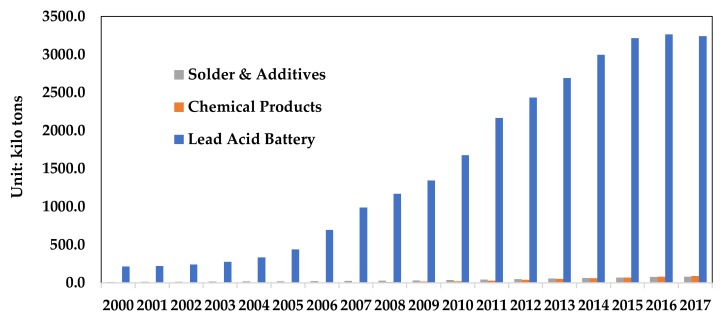
The amount of discarded lead products in China from 2000 to 2017.

**Figure 8 ijerph-17-02184-f008:**
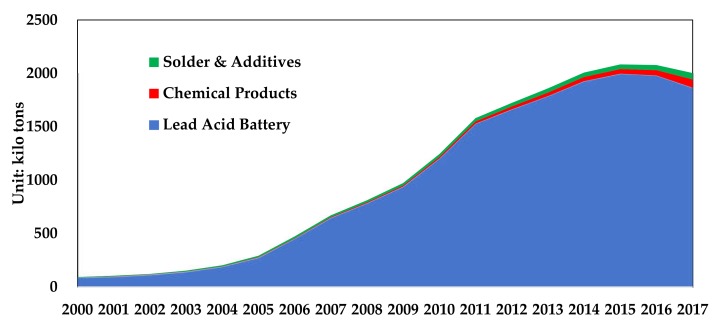
China’s recycled lead waste in 2000–2017. (Data source: [22,23]).

**Figure 9 ijerph-17-02184-f009:**
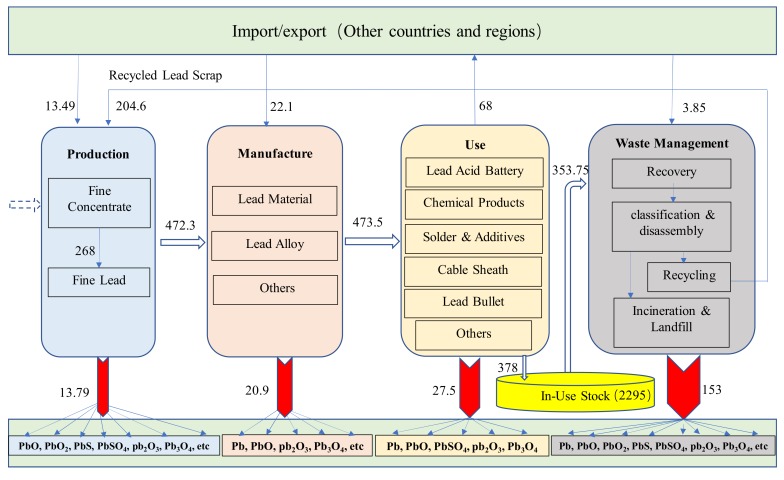
Material flow diagram of four phases of lead-bearing surface cycle in China in 2017.

**Table 1 ijerph-17-02184-t001:** Proportion of lead per unit product.

Kind	Product	Lead Factor	Setting Basis
Lead concentrate	-	52%–64%	China Nonferrous Metals Industry Yearbook
Refined lead	-	>99.9%	Lead ingots GB/T 469-2013
Lead material	Lead and lead-antimony alloy pipes, plates, rods, and wires	>99.9%	GB/T 1470-2014, GB/T 1472-2014, GB/T 1473-2014, GB/T 1474-2014, YS/T 636-2007
Lead antimony alloy		91.2%–99.1%	
Lead chemicals	Lead salt stabilizer	>82%	Tribasic lead sulfate HG/T 2340-2005, Dibasic lead phosphite HG/T 2339-2005
Other lead alloys	Cable sheath	>99%	Lead alloy ingot for cable sheath GB/T 26011-2010
	fuse	>96%	Insurance lead GB 3132-82
	Cast bearing alloy	70%–80%	Cast bearing alloy ingots GB/T 8740-2013
	Tin-based alloy	<0.35%	Methods for chemical analysis of lead-based alloys GB/T 4103-2012
Lead metal products	Lead glass	26%–30%	ASTM C1572-2004
solder	Tin-lead solder	4%–95%	GB/T 3131-2001
Lead scrap	-	>30%	GB/T 13588-2006
Lead–acid batteries	Lead content per unit mass	Approximately 70%	Research and literature data
	Lead content per product (t/kVAh)	Approximately 0.02	

Note: GB/T—Non-compulsory standards of the people’s Republic of China; YS/T—Non-ferrous metal industry standard of the people’s Republic of China; HG/T—Non-compulsory standard for chemical industry of the people’s Republic of China; ASTM C—Standard Guide for Dry Lead Glass and Oil-Filled Lead Glass Radiation Shielding Window Components for Remotely Operated Facilities; GB— National standard of the people’s Republic of China; t/kVAh—is tons per kilovolt amp-hours, represents the estimated lead content based on battery power.

**Table 2 ijerph-17-02184-t002:** Lead content in major leaded products.

Lead Concentrate	Fine Lead	Lead Material	Other Lead Alloys	Lead Antimony Alloy
55%	99.99%	95%	35%	93%–95%
Lead chemicals	Lead metal products	Lead waste and scrap	Lead–acid batteries	
			The proportion of unit mass	Lead content per product (t/kVAh)
85%	30%	80%	69.5%–73%	0.019–0.030

Note: t/kVAh—is tons per kilovolt amp-hours, represents the estimated lead content based on battery power.

**Table 3 ijerph-17-02184-t003:** The loss of various lead compounds in China’s lead smelting stage in 2017. (Unit: kilo tons).

Category	PbS	PbO	PbSO_4_	PbO_2_	Pb_2_O_3_	Pb_3_O_4_	Others
Loss Primary	23.1	30.8	9.2	----	7.8	4.6	1.5
Loss Secondary	20.7	13.2	11	15	----	----	1
Total	43.8	44	20.2	15	7.8	4.6	2.5

**Table 4 ijerph-17-02184-t004:** The loss of various lead compounds in China’s lead manufacture stage in 2017. (Unit: kilo tons).

Category	Pb	PbO	PbSO_4_	Pb_2_O_3_	Pb_3_O_4_	Others
Loss Battery	67.14	107.13	26.7	----	----	----
Loss Alloy	0.67	5.7	----	----	----	0.33
Loss Oxide	0.42	0.77	----	0.14	0.07	----
Total	68.23	11.36	26.7	0.14	0.07	0.33

**Table 5 ijerph-17-02184-t005:** The loss of various lead compounds in China’s lead use stage in 2017. (Unit: kilo tons).

Category	Pb	PbO	Pb_2_O_3_	Pb_3_O_4_	Others
Loss Chemicals	----	154.05	23.7	59.25	----
Loss Solder and additives	4.75	----	----	----	14.25
Loss Cable sheath	13.3	0.07	----	----	----
Loss Bullet	5	----	----	----	----
Total	23.05	154.12	23.7	59.25	14.25

**Table 6 ijerph-17-02184-t006:** The loss of various lead compounds in China’s lead waste management stage in 2017. (unit: kilo tons).

Category	Pb	PbO	Pb_2_O_3_	PbSO_4_	PbO_2_	PbCO_3_	Pb_3_O_4_	Others
Loss LCB	23.358	17.862	----	68.7	27.48	----	----	----
Loss S and A	----	1.205	0.192	0.131	----	0.77	0.963	0.589
Loss CH	----	7.533	1.19	0.405	----	----	1.19	1.582
Total	23.358	26.6	1.382	69.236	27.48	0.77	2.153	2.171

Note: LCB—lead–acid battery; S and A—lead solders and additives; CH—lead chemical products.

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
