# Peer review of "Environmental Pollution Effect Analysis of Lead Compounds in China Based on Life Cycle"

_ijerph, 2020, doi:10.3390/ijerph17072184_

Round 1
Reviewer 1 Report
Very nice work!
A few items:
Many places PBS instead of PbS.
Some quoted statistics need a reference, e.g., L. 51-52 WHO stats.
L 55 give level so that I don't have to go to the reference.
L 71 what is the meaning of "lead over standard rate...1.5%?"
L 74 don't use word "scum" as it also has another nasty slang meaning.
L 114 "iron imports" ?
L 124 just aboe it in table: is t/kVAh tonnes per kilovolt amp-hours?
Also in that table define all abbreviations GB/T, YS/T, etc
L 154 loss of loss
L 186 I think youmean that 95% lead is in the crude lead, not the "slag containing 95% lead."
Fig. 3 Table 3, and other Figs. Use of scale of 10 kiloton is confusing. Why not kiloton 50, 100, 150, etc.
Table 5 needs definitions of abbreviations - I assume that they are the items in Fig 6. Same Tahble 6, but is LCB actually supposed to be LAB?
L 410 References are disordered numerically.
Should probably state at beginning that your tons are metric tonnes (1000 kg) rather than English or short tons (2000 lbs) that are common and traditional in US/UK usage.
Author Response
Dear reviewer,
Thank you very much for your careful review and detailed comments on our paper. The constructive suggestions helped us greatly in our revision. Referring your comments, new paper had corrected some inaccurate description, and added clearer description than before. We have improved our paper according to your wonderful advises. Detailed responses to reviewers are showed below one by one.
Reviewer #1:
- Many places PBS instead of PbS.
- We carefully checked the whole paper and replaced “PBS” instead of “PbS” in five places.
- Some quoted statistics need a reference, e.g., L. 51-52 WHO stats.
- We have added references and their corresponding contents. “According to the World Health Organization, lead exposure contributes to about 600,000 new cases of children with intellectual disabilities every year. Aside from the poisoned futures these children suffer, the economic losses are huge: by lowering the IQ of children, lead exposure costs low and middle-income countries $977 billion per year [11].”
- L 55 give level so that I don't have to go to the reference.
- We have perfected the quoted content according to the reference [13]. “The effective control of lead production, consumption, use and recycling is an important prerequisite to reduce lead exposure. The present study found a Geometric mean value blood lead level (BLL) of 26.7μg/L in Chinese children, with 8.6% exceeding 50μg/L. However, compared to countries with a very high Human Development Index (HDI ≥ 0.9) such as Japan, Australia, France, Canada, and the US, Chinese children have much higher BLLs.”
- L 71 what is the meaning of "lead over standard rate...1.5%?"
- We changed this sentence to “In 2014, The Bulletin of China's Soil Pollution Investigation also showed that the value of lead in cultivated land over National Standard Rate was 1.5%, including lead in the main pollutants in the metal smelting industrial park and its surrounding soil.”
- L 74 don't use word "scum" as it also has another nasty slang meaning.
- We changed "scum" to "dross".
- L 114 "iron imports”?
- We changed " iron imports " to " lead imports ".
- L 124 just above it in table: is t/kVAh tons per kilovolt amp-hours?
- Yes. It represents the lead content of the unit power of the lead battery pack.
- Also, in that table define all abbreviations GB/T, YS/T, etc.
- We have defined GB/T, YS/T, GB and so on under the table.
Note: GB/T-- Non-compulsory standards of the people's Republic of China; YS/T-- Non-ferrous metal industry standard of the people's Republic of China; HG/T-- Non-compulsory standard for chemical industry of the people's Republic of China; ASTM C-- Standard Guide for Dry Lead Glass and Oil-Filled Lead Glass Radiation Shielding Window Components for Remotely Operated Facilities; GB-- National standard of the people's Republic of China; t/kVAh-- is tons per kilovolt amp-hours, represents the estimated lead content based on battery power.
- L 154 loss of loss?
- We changed " loss of loss " to " dispersed losses ".
- L 186 I think your mean that 95% lead is in the crude lead, not the "slag containing 95% lead."
- YES. We changed this sentence. “The grade of lead concentrate is about 60%, and the lead concentrate is sent to the smelting furnace to produce the crude lead containing about 95% lead.”
- Fig. 3 Table 3, and other Figs. Use of scale of 10 kiloton is confusing. Why not kiloton 50, 100, 150, etc.
- We have changed the units of all the charts and watches from 10 kilotons to kilotons.
- Table 5 needs definitions of abbreviations - I assume that they are the items in Fig 6. Same Table 6, but is LCB actually supposed to be LAB?
- We have defined LCB, S&A, CH under the Table 6. “Note: LCB-- lead-acid battery; S&A—lead solders and additives; CH-- lead chemical products.”
- L 410 References are disordered numerically.
- We have revised the reference number.
- Should probably state at beginning that your tons are metric tonnes (1000 kg) rather than English or short tons (2000 lbs) that are common and traditional in US/UK usage.
- We have already made the state of unit tons in Section 2.4. “In this paper the tons are metric tons (1000 kg).”
Reviewer 2 Report
In abstract, line 21 you have written PBS, instead PbS. Again in line 24 is written PBS. In line 93 agian appears this (PBS), are you describing lead sulfur? or it is another description?
The paper Environmental pollution effect analysis of lead compounds in China based on life cycle, is related to the evaluation of the lead life cycle using a material flow analysis method, which is interesting.
I this this paper has good information and could be improved takimng in account the following
Authors could to describe in more detail the novelty of paper in relation with the direct effect of the lead loss for the human health and environment.
Also, I think this paper is quiet regional, but the method could be used to stimate its use for other kind of hazardous materials and even in other regions. If authors can improve this, paper will be improved.
Another observations
Line 195, check the number 0.850 kilo tons, you have a mistake.
Line 200, you have called the PBS as sulfide, then for my knowledgement this must be written PbS.
Line 204, yu have written PBO instead PbO. Please ceck and correct it
In line 46, the word utilization appears to be duplicated. Please check this point.
Please check labels of figure 4.
Line 382, formula has no subscripts
Author Response
Response to reviewer
Dear reviewer,
Thank you very much for your careful review and detailed comments on our paper. The constructive suggestions helped us greatly in our revision. Referring your comments, new paper had corrected some inaccurate description, and added clearer description than before. We have improved our paper according to your wonderful advises. Detailed responses to reviewers are showed below one by one.
Reviewer #2:
- In abstract, line 21 you have written PBS, instead PbS. Again, in line 24 is written PBS. In line 93 again appears this (PBS), are you describing lead sulfur? or it is another description?
- We carefully checked the whole paper and replaced “PBS” instead of “PbS” in five places.
- Authors could to describe in more detail the novelty of paper in relation with
- We have added some content that the direct effect of the lead loss for the human health and environment. “Lead (Pb) is a naturally occurring element found in the earth’s crust with various uses largely owing to its malleability and corrosion resistance. Multiple uses of Pb during the 20th century, including leaded petrol, lead-based paint and solder in water pipes, have resulted in elevated Pb exposure risk to humans. Young children, in particular, are more vulnerable to Pb exposure due to their frequent hand-to-mouth activity, higher absorption rates and their developing central nervous system [1].”
- Also, I think this paper is quiet regional, but the method could be used to stimate its use for other kind of hazardous materials and even in other regions. If authors can improve this, paper will be improved.
- We have added some content to the introduction section. “Since the 21st session of the United Nations Environment Program in 2001, a series of resolutions on lead control have been formed to promote global action to control the environmental exposure and health risk of lead. China's continuous measures are effective, for example, in the areas of mining, metal smelting and production, lead emissions and environmental impact have declined significantly. The continuous decline of children's blood lead index is an important proof. The focus of lead pollution has gradually shifted to the use and scrap management of urban lead products. The existing research results show that the distribution of blood lead index is unbalanced in all provinces of China [13]. This requires a new perspective to assess the environmental impact of lead metabolism. Material flow analysis is a systematic evaluation tool, which can characterize the flow, and flow dynamic changes of lead and its compounds in life cycle. This allows us to grasp the impact of lead on the environment in the whole life cycle and assess the health risk. Help us to identify the direction of environmental management of urban lead and other hazardous products. Provide decision support for the sustainable development of urban health.”
- We have added some content to the conclusion section.
“(5) The research on the environmental impact of lead and other toxic metals should be changed from the national level to the regional level, especially the urban level. The results of this study have shown that the environmental impact of lead treatment effect is significant. The focus of lead pollution control has changed from "flow" management of industrial production to "stock" of social consumption. City is the accumulation center of social consumer goods. The use and discard management of toxic metal products has become the main content of urban environmental improvement and health management. Quantitative analysis of the geographical distribution of toxic metal products, as well as waste recycling and treatment of pollution. It is of great significance to provide improvement and Governance Space for urban environment and health management.
(6) The research methods and ideas of this paper can also be extended to the quantitative analysis of environmental impact of heavy metals such as cadmium, mercury, copper, nickel, zinc and manganese. Through the collaborative and coupling research with urban water resources, the impact of heavy metal products on water, atmosphere and soil in the process of using dissipation and waste incineration and landfill was analyzed. Promote strict "footprint" management of heavy metal products in cities. This has a very good supporting role for the comprehensive management and sustainable development of urban environment.”
- Line 195, check the number 0.850 kilo tons, you have a mistake.
- We have changed the unit to “million”.
- Line 200, you have called the PBS as sulfide, then for my knowledge this must be written PbS.
- We have changed “PBS” to “PbS”.
- Line 204, you have written PBO instead PbO. Please check and correct it.
- We have changed “PBO” to “PbO”.
- In line 46, the word utilization appears to be duplicated. Please check this point.
- We changed this sentence. “This can help humans to identify and predict the flow and direction of specific substances, and scientifically analyze the efficiency of human resources utilization.”
- Please check labels of Figure 4.
- We have changed Figure 4.
- Line 382, formula has no subscripts.
- We modify the parameter subscript in the parameter explanation part of Equations (8). “Xi, Ri “